# Tree of Clarifications: Answering Ambiguous Questions with Retrieval-Augmented Large Language Models

**Gangwoo Kim**[1] **Sungdong Kim**[2,3,4] **Byeongguk Jeon**[1] **Joonsuk Park**[2,3,5] **Jaewoo Kang**[1†]

Korea University[1]    NAVER Cloud[2]    NAVER AI Lab[3]
KAIST AI[4]    University of Richmond[5]
{gangwoo_kim, bkjeon1211, kangj}@korea.ac.kr
sungdong.kim@navercorp.com    park@joonsuk.org

## Abstract

Questions in open-domain question answering are often ambiguous, allowing multiple interpretations. One approach to handling them is to identify all possible interpretations of the ambiguous question (AQ) and to generate a long-form answer addressing them all, as suggested by Stelmakh et al. (2022). While it provides a comprehensive response without bothering the user for clarification, considering multiple dimensions of ambiguity and gathering corresponding knowledge remains a challenge. To cope with the challenge, we propose a novel framework, TREE OF CLARIFICATIONS (TOC): It recursively constructs a tree of disambiguations for the AQ—via few-shot prompting leveraging external knowledge—and uses it to generate a long-form answer. ToC outperforms existing baselines on ASQA in a few-shot setup across all metrics, while surpassing fully-supervised baselines trained on the whole training set in terms of Disambig-F1 and Disambig-ROUGE. Code is available at github.com/gankim/tree-of-clarifications.

## 1 Introduction

In open-domain question answering (ODQA), users often ask ambiguous questions (AQs), which can be interpreted in multiple ways. To handle AQs, several approaches have been proposed, such as providing individual answers to disambiguated questions (DQs) for all plausible interpretations of the given AQ (Min et al., 2020) or asking a clarification question (Guo et al., 2021). Among them, we adopt that of Stelmakh et al. (2022), which provides a comprehensive response without bothering the user for clarification: The task is to identify all DQs of the given AQ and generate a long-form answer addressing all the DQs (See Figure 1).

There are two main challenges to this task: (1) the AQ may need to be clarified by considering mul-

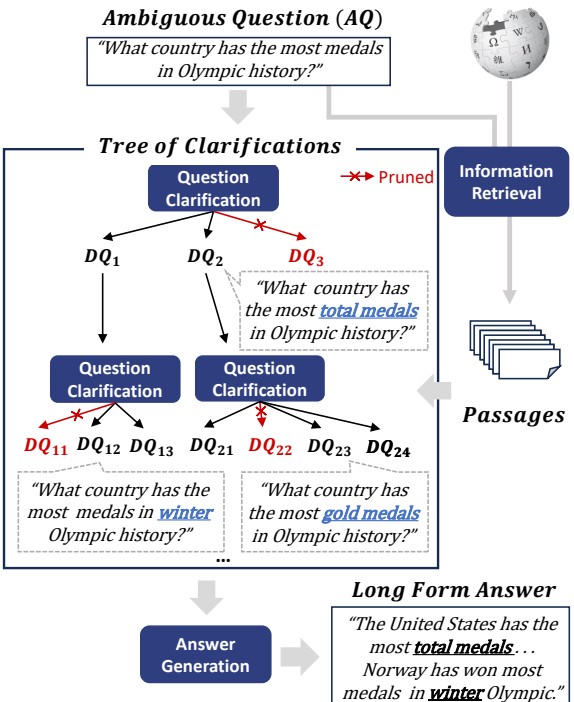

Figure 1: Overview of TREE OF CLARIFICATIONS. (1) relevant passages for the ambiguous question (AQ) are retrieved. (2) leveraging the passages, disambiguated questions (DQs) for the AQ are recursively generated via few-shot prompting and pruned as necessary. (3) a long-form answer addressing all DQs is generated.

tiple dimensions of ambiguity. For example, the AQ *what country has the most medals in Olympic history* in Figure 1 can be clarified with respect to the type of medals—gold, silver, or bronze—or Olympics—summer or winter; and (2) substantial knowledge is required to identify DQs and respective answers. For example, it requires knowledge to be aware of the existence of different types of medals and the exact counts for each country.

To address the challenges and provide a long-form answer to AQ, we propose a novel framework, TREE OF CLARIFICATIONS (TOC): It recursively constructs a tree of DQs for the AQ—via few-shot

---

† Corresponding author

prompting leveraging external knowledge—and uses it to generate a long-form answer. More specifically, first, relevant passages for the AQ are retrieved. Then, leveraging the passages, DQs for the AQ are recursively generated via few-shot prompting and pruned as necessary. Lastly, a long-form answer addressing all DQs is generated. The tree structure promotes exploring DQs in targeting particular dimensions of clarification, addressing the first challenge, and the external sources offer additional knowledge to cope with the second challenge.

Experiments demonstrate that our proposed use of LLMs with retrieval-augmentation and guidance to pursue diverse paths of clarification results in the new state-of-the-art on ASQA (Stelmakh et al., 2022)—a long-form QA benchmark for AQs. ToC outperforms existing baselines on ASQA in a few-shot setup across all metrics. In addition, this 5-shot performance surpasses that of the fully-supervised baselines trained on the whole training set by 7.3 and 2.9 in terms of Disambig-F1 and Disambig-ROUGE, respectively.

The main contribution of this work is proposing a novel framework, TREE OF CLARIFICATIONS (ToC), for generating long-form answers to AQs in ODQA, advancing the state-of-the-art on the ASQA benchmark. ToC introduces two main innovations:

- It guides LLMs to explore diverse paths of clarification of the given AQ in a tree structure with the ability to prune unhelpful DQs.

- To the best of our knowledge, it is the first to combine retrieval systems with LLM for generating long-form answers to AQs.

## 2  Related Work

A line of studies (Min et al., 2020, 2021; Gao et al., 2021; Shao and Huang, 2022) extends retrieve-and-read frameworks dominant in ODQA task (Chen et al., 2017; Karpukhin et al., 2020; Lewis et al., 2020; Izacard and Grave, 2021) to clarify AQ and generate DQs with corresponding answers to them. However, their approaches require fine-tuning models on the large-scale train set. On the other hand, our framework enables LLM to generate a comprehensive response addressing all DQs via few-shot prompting.

Recent studies introduce LLM-based methods to generate a long-form answer to the AQ. Amplayo

et al. (2023) suggest optimal prompts specifically engineered for the task. Kuhn et al. (2022) prompt LLMs to clarify ambiguous questions selectively. However, the studies do not utilize external information to ensure the factual correctness of the disambiguations, thereby potentially increasing the risk of hallucinations from LLMs. Moreover, the results could be bounded by inherent parametric knowledge of LLM. Concurrently, Lee et al. (2023) automatically generate clarifying questions to resolve ambiguity.

Our framework involves the recursive tree architecture, inspired by several prior studies. Min et al. (2021) propose the tree-decoding algorithm to autoregressively rerank passages in ambiguous QA. Gao et al. (2021) iteratively explore additional interpretations and verify them in a round-trip manner. Concurrently, extending chain of thoughts (Wei et al., 2022) prompting, Yao et al. (2023) apply the tree architecture to reasoning tasks for deductive or mathematical problems. On the contrary, ToC recursively clarifies questions and introduces a self-verification method to prune unhelpful DQs.

## 3  Tree of Clarifications

We introduce a novel framework, TREE OF CLARIFICATIONS (ToC), as illustrated in Figure 1. We first devise retrieval-augmented clarification (RAC; Sec. 3.1), a basic component that clarifies AQ and generates DQs based on relevant passages. ToC explores various fine-grained interpretations, represented as a tree structure (TS; Sec. 3.2) by recursively performing RAC and pruning unhelpful DQs. Lastly, it aggregates the tree and generates a long-form answer addressing all valid interpretations.

### 3.1  Retrieval-Augmented Clarification (RAC)

We first retrieve relevant Wikipedia documents for the AQ by using two retrieval systems, ColBERT (Khattab and Zaharia, 2020) and Bing search engine[1]. ColBERT is a recent dense retriever that has effective and efficient zero-shot search quality. Following Khattab et al. (2022), we use the off-the-shelf model pre-trained on MS-Marco (Bajaj et al., 2016). We additionally include the Bing search engine to promote the diversity of retrieved Wikipedia passages. Finally, we obtain over 200 passages by combining passages retrieved by each system.

---

[1] https://www.microsoft.com/bing

After collecting a passage set for the AQ, we rerank and choose top-$k$ passages and augment them to a prompt. We use SentenceBERT (Reimers and Gurevych, 2019) pre-trained on MS-Marco as the reranker backbone. For in-context learning setup, we dynamically choose $k$-shot examples with the nearest neighbor search[2] and add them to the prompt. We initiate with the instruction of Amplayo et al. (2023) and revise it for our setup. Given the prompt with relevant passages and AQs, LLM generates all possible DQs and their corresponding answers[3].

## 3.2 Tree Structure (TS)

To effectively explore the diverse dimensions of ambiguity, we introduce a recursive tree structure of clarifications. Starting from the root node with AQ, it progressively inserts child nodes by recursively performing RAC, each of which contains a disambiguated question-answer pair. In each expansion step, passages are reranked again regarding the current query. It allows each step to focus on its own DQ, encouraging ToC to comprehend a wider range of knowledge. Exploration of a tree ends when it satisfies termination conditions; it reaches the maximum number of valid nodes or the maximum depth. We choose the breadth-first search (BFS) by default, hence the resulting tree could cover the broader interpretations[4].

**Pruning with Self-Verification**    To remove unhelpful nodes, we design a pruning method, inspired by current studies for self-verification (Kadavath et al., 2022; Cole et al., 2023). Specifically, we check the factual coherency between the answers in a target node and the AQ in the root node. By doing so, we discard the generated DQs that ask different or irrelevant facts from the original one. For example, given an AQ "*Who will host the next world cup 2022?*", a generated disambiguation "*DQ: Who hosted the world cup 2018? A: Russia*" is a factually consistent question-answer pair but it changes the original scope of the AQ[5]. We perform self-verification by prompting LLMs to determine whether the current node would be pruned or not. Prompted with AQ, the answer to the target DQ, and the answer-containing passage, LLM identifies

---

[2]See Appendix A.3 for detailed implementation

[3]See Appendix C.2 for example prompts

[4]It is suboptimal to adopt the depth-first search since it would encounter unambiguous questions more frequently. See Appendix 7 for failure cases.

[5]See Appendix C.3 for more detailed case studies

| Model | D-F1 | R-L | DR |
|---|---|---|---|
| *Fully-supervised* | | | |
| T5-Large Closed-Book | 7.4 | 33.5 | 15.7 |
| T5-Large w/ JPR | 26.4 | 43.0 | 33.7 |
| PaLM w/ Soft Prompt Tuning* | 27.8 | 37.4 | 32.1 |
| *Few-shot Prompting (5-shot)* | | | |
| PaLM* | 25.3 | 34.5 | 29.6 |
| GPT-3* | 25.0 | 31.8 | 28.2 |
| *Tree of Clarifications (ToC; Ours)* | | | |
| GPT-3 + RAC | 31.1 | 39.6 | 35.1 |
| GPT-3 + RAC + TS | 32.4 | **40.0** | 36.0 |
| GPT-3 + RAC + TS w/ Pruning | **33.7** | 39.7 | **36.6** |

\* from Amplayo et al. (2023)

Table 1: Evaluation results for long-form QA on ambiguous questions from the development set of ASQA (Stelmakh et al., 2022). Baselines are either fully-supervised or 5-shot prompted. Note, ToC framework consists of retrieval-augmented clarification (RAC) and tree structure (TS).

if the given answer could be a correct answer to AQ.

**Answer Generation**    Once constructing the tree of clarifications, ToC aggregates all valid nodes and generates a comprehensive long-form answer to AQ. It selects the disambiguations in retained nodes of the resulting tree with the relevant passages. If the number of nodes is insufficient, we undo the pruning steps from closer nodes to the root node in BFS order. Passages that contain the answers of valid nodes are prioritized. It finally generates a long-form answer, encoding AQ, selected disambiguations, and relevant passages[6].

## 4 Experiment

### 4.1 Experimental Setup

**Datasets**    All baselines and our framework are evaluated on ASQA (Stelmakh et al., 2022). It is a long-form QA dataset built upon the 6K ambiguous questions identified from AmbigNQ (Min et al., 2020). More details are in Appendix A.1

**Evaluation Metrics**    We use three evaluation metrics, following Stelmakh et al. (2022). (1) **Disambig-F1 (D-F1)** measures the factual correctness of generated predictions. It extracts short answers to each DQ and computes their F1 accuracy. (2) **ROUGE-L (R-L)** measures the lexical overlap between long-form answers from references and predictions. (3) **DR** score is the geometric mean of

---

[6]See Appendix C.4 for an example prompt

| Model | D-F1 | R-L | DR |
|---|---|---|---|
| GPT-3 (Baseline) | 24.2 | 36.0 | 29.5 |
| GPT-3 w/ RAC | **31.1** | **39.6** | **35.1** |
| − Disambiguations | 30.5 | 37.3 | 33.7 |
| − Bing Search Engine | 28.5 | 37.4 | 32.7 |
| − Retrieval Systems | 25.6 | 35.1 | 30.0 |

Table 2: Ablation study on all components of retrieval-augmented clarification (RAC).

| Filtration | #(DQs) | Answer-F1 |
|---|---|---|
| w/o Pruning (None) | 12,838 | 40.9 |
| w Pruning | | |
| + Deduplication | 10,598 | 40.1 |
| + Self-Verification | 4,239 | 59.3 |

Table 3: Ablated results with and without pruning methods. The number of retained DQs after pruning and Answer-F1 are reported.

two scores, which assesses the overall performance. For validating intermediate nodes, we additionally use **Answer-F1** that measures the accuracy of generated short answers in disambiguation. Further details are in Appendix A.2.

**Baselines** Stelmakh et al. (2022) propose fine-tuned baselines. They fine-tune T5-large (Raffel et al., 2020) to generate long-form answers on the whole train set. Models are evaluated in the closed-book setup or combined with JPR (Min et al., 2021), task-specific dense retriever for ambiguous QA by enhancing DPR (Karpukhin et al., 2020). On the other hand, Amplayo et al. (2023) propose a prompt engineering method to adapt LLMs to the ASQA benchmark. They employ PaLM (Chowdhery et al., 2022) and Instruct-GPT (Ouyang et al., 2022) that learn the soft prompts or adopt in-context learning with few-shot examples. They conduct experiments in the closed-book setup. Note that they share the same backbone with our models, GPT-3 with 175B parameters (text-davinci-002).

## 4.2 Experimental Results

**TOC outperforms fully-supervised and few-shot prompting baselines.** Table 1 shows the long-form QA performance of baselines and TOC on the development set of ASQA. Among baselines, using the whole training set (*Fully-supervised*) achieves greater performances than *Few-shot Prompting* in all metrics. It implies that long-form QA task is challenging in the few-shot setup. In the closed-book setup, GPT-3 shows competitive performances with T5-large with JPR in D-F1 score, showing LLM's strong reasoning ability over its inherent knowledge.

Among our models, LLM with RAC outperforms all other baselines in D-F1 and DR scores. It indicates the importance of leveraging external knowledge in clarifying AQs. Employing the tree structure (TS) helps the model to explore diverse interpretations, improving D-F1 and DR scores by

1.3 and 0.9. When pruning the tree with our proposed self-verification (TS w/ Pruning), the model achieves state-of-the-art performance in D-F1 and DR score, surpassing the previous few-shot baseline by 8.4 and 7.0. Notably, it outperforms the best model in a fully-supervised setup (T5-large with JPR) by 7.3 and 2.9. In the experiment, T5-Large in a closed-book setup achieves comparable performance with LLM baselines in ROUGE-L score despite its poor D-F1 scores. It reconfirms the observation from Krishna et al. (2021) that shows the limitations of the ROUGE-L metric.

**Integrating retrieval systems largely contributes to accurate and diverse disambiguations.** Table 2 displays the ablation study for measuring the contributions of each proposed component. When removing disambiguations from few-shot training examples, the ROUGE-L score is significantly degraded, which shows the importance of the intermediate step to provide the complete answer. Integrating retrieval systems (i.e., Bing search engine and ColBERT) largely improves the model performance, especially in the D-F1 score. It indicates using external knowledge is key to enhancing the factual correctness of clarification. We report intrinsic evaluation for each retrieval system in Appendix B.

**Our pruning method precisely identifies helpful disambiguations from the tree.** Table 3 shows intrinsic evaluation for generated disambiguations, where all baselines are evaluated with Answer-F1 score that measures the F1 accuracy of the answer to the target DQ. Compared to the baseline, the valid nodes that pass self-verification contain more accurate disambiguations, achieving much higher Answer-F1 score (+18.4). On the other hand, solely using deduplication does not advance the accuracy, indicating the efficacy of our proposed self-verification method.

## 5   Discussion

**Ambiguity Detection**   ToC is designed to clarify AQs without bothering users; hence does not explicitly identify whether the given question is ambiguous or not. It tries to perform clarification even if the question cannot be disambiguated anymore, often resulting in generating duplicate or irrelevant DQs[7]. However, we could presume a question to be unambiguous if it can no longer be disambiguated[8]. In ToC, when it fails to disambiguate the given question or all generated disambiguations are pruned, the question could be regarded as unambiguous.

**Computational Complexity**   Although ToC requires multiple LLM calls, its maximum number is less than 20 times per question. Exploration of the tree ends when it obtains the pre-defined number of valid nodes (10 in our experiments). Since the clarification process generates from two to five disambiguations for each question, it satisfies the termination condition in a few steps without the pruning method. Failing to expand three times in a row also terminates the exploration. Pruning steps consume a smaller amount of tokens since they encode a single passage without few-shot exemplars. Compared to the existing ensemble methods such as self-consistency (Wei et al., 2022) which cannot be directly adopted to the generative task, ToC achieves a state-of-the-art performance with a comparable number of LLM calls.

**Generalizability**   The key idea of ToC could be potentially generalized to other tasks and model architectures. It has a model-agnostic structure that could effectively explore diverse paths of recursive reasoning, which would be helpful for tasks that require multi-step reasoning, such as multi-hop QA. Future work might investigate the generalizability of ToC to diverse tasks, datasets, and LM architectures.

## 6   Conclusion

In this work, we propose a novel framework, TREE OF CLARIFICATIONS. It recursively builds a tree of disambiguations for the AQ via few-shot prompting with external knowledge and utilizes it to generate a long-form answer. Our framework explores diverse dimensions of interpretations of ambiguity. Experimental results demonstrate ToC successfully guide LLMs to traverse diverse paths of clarification for a given AQ within tree structure and generate comprehensive answers. We hope this work could shed light on building robust clarification models, which can be generalized toward real-world scenarios.

## Limitations

Although ToC is a model-agnostic framework that could be combined with other components, our study is limited in demonstrating the generalizability of different kinds or sizes of LLMs. In addition, the experiments are only conducted on a benchmark, ASQA (Stelmakh et al., 2022). Although ToC enables LLM to explore diverse reasoning paths by iteratively prompting LLM, the cost of multiple prompting is not negligible.

We tried the recent prompting method, chain of thoughts (Wei et al., 2022), but failed to enhance the performance in our pilot experiments. It might indicate the disambiguation process requires external knowledge, which shows the importance of document-grounded or retrieval-augmented systems. Future work could suggest other pruning methods that identify unhelpful DQs more effectively. The performance could be further enhanced by using the state-of-the-art reranker in the answer sentence selection task, as proposed by recent works (Garg et al., 2020; Lauriola and Moschitti, 2021).

## Acknowledgements

The first author, Gangwoo Kim, has been supported by the Hyundai Motor Chung Mong-Koo Foundation. This research was supported by the National Research Foundation of Korea (NRF-2023R1A2C3004176, RS-2023-00262002), the MSIT (Ministry of Science and ICT), Korea, under the ICT Creative Consilience program (IITP-2022-2020-0-01819) supervised by the IITP (Institute for Information & communications Technology Planning & Evaluation), and the Electronics and Telecommunications Research Institute (RS-2023-00220195).

---

[7]See Appendix 7 for failure cases

[8]The idea is aligned with the annotation process of AmbigQA (Min et al., 2020), in which the target question is classified as ambiguous if multiple distinct answers to it were observed.

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

## A   Experimental Setup Details

### A.1   Ambiguous QA Datasets

All baselines and our framework are evaluated on ASQA benchmark (Stelmakh et al., 2022). It is a long-form QA dataset built on the subset of ambiguous questions identified from AmbigNQ dataset (Min et al., 2020). It contains open-domain questions collected from Natural Questions (Kwiatkowski et al., 2019). ASQA consists of 6,316 ambiguous questions and their long-form answers with disambiguations, split into 4,353, 948, and 1,015 train, development, and test set, respectively.

### A.2   Evaluation Metrics

Following Stelmakh et al. (2022), we use three evaluation metrics on ASQA. First, **ROUGE-L (R-L)** measures the lexical overlap between long-form answers from references and system-generated predictions. Since the benchmark provides two ground-truth answers, we report the maximum ROUGE-L score. **Disambig-F1 (D-F1)** measures the factual correctness of generated predictions. A reading comprehension model, RoBERTa (Liu et al., 2019) trained on SQuADv2 (Rajpurkar et al., 2018), finds short answers to the ground-truth DQs from the generated long-form response. Then, F1 accuracy of the detected answer is calculated to check if the long-form answer contains accurate information. **Disambiguation-ROUGE (DR)** score is computed as the geometric mean of ROUGE-L and Disambig-F1 to measure the overall performance. We additionally use **Answer-F1** to validate the disambiguations. It computes the maximum F1 accuracy of answers to a single DQ. We use ground-truth disambiguations provided by AmbigNQ (Min et al., 2020).

### A.3   Implementation Details

A large portion of our implementation is based on the DSP library (Khattab et al., 2022). To dynamically find few-shot examples with the nearest neighbor search, we use pre-trained MiniLM (Wang et al., 2020) to obtain hidden representations of questions and compute similarity scores with Faiss library (Johnson et al., 2019). We add 5-shot training examples to the prompt, following Amplayo et al. (2023). It was the optimal number in our pilot experiment.

For prompting LLM to perform RAC, we use top-5 relevant passages. To determine whether to

| Retrieval System | AC@10 | AC@30 | AC@100 |
|---|---|---|---|
| ColBERTv2 | 56.4 | 68.4 | 73.4 |
|   w/ Reranker | 56.8 | 69.0 | 73.4 |
| Bing Search Engine | 43.3 | 58.3 | 73.5 |
|   w/ Reranker | 62.7 | 68.0 | 72.8 |
| Combined w/ Reranker | **64.2** | **77.4** | **80.1** |

Table 4: Intrinsic evaluation of retrieval systems. Answer coverage at $k$ (AC@$k$) measures the proportion of disambiguated answers that are covered by top-$k$ retrieved passages.

prune the target node or not, we rerank and pick the most relevant passage among those containing the answer in the target node. In the answer generation process, we took ten valid disambiguations in BFS order and five answer-containing passages. We use API served by OpenAI[9] to employ GPT-3 as our backbone. We set max tokens as 300 and top-$p$ as 1.0.

## B   Additional Experiment

### B.1   Intrinsic Evaluation for Retrieval Systems

We randomly sample 100 examples from ASQA dataset and report intrinsic evaluation results for retrieval systems. Since a single AQ has multiple ground-truth DQs and their answers, it is not trivial to check how many answers are covered by retrieved passages. Inspired by Min et al. (2021), we devise an evaluation proxy, answer coverage, for measuring the quality of retrieved passages in ambiguous QA tasks. We consider the retrieval as successful if the retrieved passages contain one of the answers to the target DQ. We calculate the proportion of success among DQs for a single AQ to check overall answer coverage.

Table 4 compares retrieval systems in answer coverage (AC@$k$) of top-$k$ passages. Bing search engine without reranker performs worst among baselines in AC@10 and @30. However, with reranker, its performances are greatly enhanced, outperforming ColBERT baselines. When combining two retrieval systems, it shows the best performances across all evaluation metrics; hence two results are complementary. It achieves 80.1 in AC@100 scores, which indicates the passage set has sufficient information if properly explored.

---

[9]https://platform.openai.com/docs/api-reference

## C Qualitative Analysis

### C.1 Prompt Format

We add format descriptions to our prompt following Khattab et al. (2022). Table 5 displays the format specifically designed to generate disambiguations for a given question based on external documents. The format description is augmented to prompts of both RAC and the answer generation. By using it, we encouarge LLM to comply with the format.

### C.2 Question Clarification

Table 6 shows an example of RAC for the AQ. Retrieval systems provide the external knowledge. Leveraging it, LLM generates disambiguated question-answer pairs. In RAC, long-form answers are also generated to follow the format but we do not use them in the later steps.

In Table 7, we observe the cases where TOC encounters unambiguous questions and fails to clarify them. It often asks different or irrelevant facts from them of original AQ.

### C.3 Self Verification

Table 8, 9 show examples of self-verification prompt. We prompt LLM to verify the current answer is factually coherent with AQ based on the relevant passage. It generates 'True' or 'False' to determine whether the node would be discarded or not. We do not provide few-shot training examples or formats.

### C.4 Answer Generation

Table 10 depicts an example of answer generation prompt. We use a similar prompt to that of RAC except disambiguations are given as inputs. It encodes up to ten disambiguations and five relevant passages.

Follow the following format.

**Context**:
${sources that may contain relevant content}

**Question**: ${ambiguous question to be disambiguated}

**Disambiguations**: ${the disambiguated pairs of questions and answers, each is separated by a new line.}
DQ i: ${(i)-th disambiguated question that clarifies the ambiguous question}
DA i: ${short factoid answers separated by semi-colon (;) to (i)-th disambiguated question, often between 1 and 5 words}

**Answer**: ${a thorough, detailed answer that explains the multiple interpretations of the original question and includes the appropriate disambiguations, at least three sentences.}

Table 5: Format description for both RAC and the answer generation.

I will provide ambiguous questions that can have multiple answers based on their different possible interpretations. Clarify the given question into several disambiguated questions and provide short factoid answers to each question. Subsequently, summarize them into a detailed long-form answer of at least three sentences. Here are some examples.

**Context**:
[1] Fred and George Weasley | Fred and George Weasley are fictional characters in the "Harry Potter" book series written by J. K. Rowling. The characters are the identical twin brothers ... The twins were played by identical twin brothers James and Oliver Phelps in the film adaptations
[2] James and Oliver Phelps | James Andrew Eric Phelps and Oliver Martyn John Phelps (born 25 February 1986) are English actors and identical twin brothers. They are known for playing Fred and George Weasley in the "Harry Potter" film series from 2001 to 2011 ...
· · ·
[5] Chris Rankin | plays of "Bugsy Malone" and "The Lion, The Witch and The Wardrobe". His professional acting career began when he won the role of Percy Weasley in September 2000 ... after his character's absence from "Harry Potter and the Goblet"

**Question**: Who played the weasley brothers in harry potter?

**Disambiguations**:
DQ 1: Who played the fictional characters Fred and George Weasley in the "Harry Potter" book series?
DA 1: James and Oliver Phelps
DQ 2: Who are the English actors and identical twin brothers known for playing Fred and George Weasley in the "Harry Potter" film series?
DA 2: James and Oliver Phelps
DQ 3: Who is the actor that plays Percy Weasley in the Harry Potter series?
DA 3: Chris Rankin

**Answer**: The Weasley brothers in the Harry Potter series were played by identical twin brothers James and other Oliver Phelps. The Phelps brothers are English actors who have appeared in films and TV shows together as a duo. Chris Rankin is the actor who played Percy Weasley in the Harry Potter series. Rankin is a British actor who has appeared in theatre, film, and TV.

Table 6: Example prompt and output in RAC. Few-shot training examples and format descriptions are omitted for simplicity.

I will provide ambiguous questions that can have multiple answers based on their different possible interpretations. Clarify the given question into several disambiguated questions and provide short factoid answers to each question. Subsequently, summarize them into a detailed long-form answer of at least three sentences. Here are some examples.

Follow the following format.
**Context**:
[1] 1991 Major League Baseball All-Star Game | The 1991 Major League Baseball All-Star Game ...
· · ·
[5] Venues of the 1996 Summer Olympics | would serve as host to the Peach Bowl from ...

**Question**: When did Toronto host the MLB All-Star Game in 1991?

**Disambiguations**:
```
DQ 1: When was the 1991 Major League Baseball All-Star Game played?
DA 1: July 9, 1991
DQ 2: What was the outcome of the 1991 Major League Baseball All-Star Game?
DA 2: American League defeated the National League

Answer: The 1991 Major League Baseball All-Star Game was ...
```

---

I will provide ambiguous questions that can have multiple answers based on their different possible interpretations. Clarify the given question into several disambiguated questions and provide short factoid answers to each question. Subsequently, summarize them into a detailed long-form answer of at least three sentences. Here are some examples.

Follow the following format.
**Context**:
[1] Highest-paid NBA players by season | Highest-paid NBA players by season The highest-paid NBA players by season over ...
· · ·
[5] Highest-paid NBA players by season | Highest-paid NBA players ...

**Question**: Who was the highest-paid NBA player in the 2017-2018 season?

**Disambiguations**:
```
DQ 1: Who was the highest-paid NBA player in the 2017-2018 season by salary?
DA 1: LeBron James
DQ 2: Who was the highest-paid NBA player in the 2017-2018 season by total earnings?
DA 2: LeBron James

Answer: LeBron James was the highest-paid NBA player in the 2017-2018 season ...
```

Table 7: Failure case where the model encounters and clarifies unambiguous questions. Few-shot training examples and format descriptions are omitted for simplicity.

---

**Correct Case 1**

**DQ**: Who was selected to host the 2018 FIFA World Cup?

---

I will provide a question, relevant context, and proposed answer to it. Identify whether the proposed answer could be correct answers or not with only 'True' or 'False'

**Context**:

2018 and 2022 FIFA World Cup bids | FIFA's headquarters in Zurich. Russia was chosen to host the 2018 World Cup, and Qatar was chosen to host the 2022 World Cup. This made Russia the first Eastern European country to host the World Cup, while Qatar would be the first Middle Eastern country to host the World Cup. Blatter noted that the committee had decided to "go to new lands" and reflected a desire to "develop football" by bringing it to more countries. In each round a majority of twelve votes was needed. If no bid received 12 votes in a round, the bid with the fewest votes

**Question**: Who is hosting the next world cup 2022?

**Proposed Answer**: Russia

`False`

---

**Correct Case 2**

**DQ**: Which player has won the most World Series in baseball?

---

I will provide a question, relevant context, and proposed answer to it. Identify whether the proposed answer could be correct answers or not with only 'True' or 'False'

**Context**:

World Series ring | on World Series rings. The New York Yankees Museum, located in Yankee Stadium, has an exhibit with replicas of all Yankees' World Series rings, including the pocket watch given after the 1923 World Series. Yogi Berra won the most World Series rings with 10, as a player. Frankie Crosetti won 17 as a player and as a coach. Yogi Berra Museum and Learning Center. World Series ring A World Series ring is an award given to Major League Baseball players who win the World Series. Since only one Commissioner's Trophy is awarded to the team, a World Series ring is

**Question**: Who's won the most world series in baseball?

**Proposed Answer**: Yogi Berra

`True`

---

Table 8: Correct cases of pruning method. Few-shot training examples or formats are not augmented to the prompt. Generated texts are colored green.

**Incorrect Case 1**

**DQ**: Who is the highest goalscorer in world football in a single game?

I will provide a question, relevant context, and proposed answer to it. Identify whether the proposed answer could be correct answers or not with only 'True' or 'False'

**Context**:

List of footballers with the most goals in a single game | This is a list of players with the most goals in a football game. The list only includes players who have scored the most multiple goals in first class or fully professional matches for country or club. The current world record for an international is held by Archie Thompson, who scored 13 goals against American Samoa in Australia's 31–0 victory during the 2002 FIFA World Cup qualification. David Zdrilic scored 8 goals. In November 2022, Shokhan Nooraldin Salihi scored 15 goals in the match of Al-Hilal against Sama in the 2022–23 Saudi Women's Premier League. In this match, Al-Hilal beat Sama 18-0.

**Question**: Who has the highest goals in world football?

**Proposed Answer**: Archie Thompson

False

**Incorrect Case 2**

**DQ**: When was episode 113 of Dragon Ball Super released in the US?

I will provide a question, relevant context, and proposed answer to it. Identify whether the proposed answer could be correct answers or not with only 'True' or 'False'

**Context**:

Dragon Ball Super | would be available in the United States in summer 2017. Bandai has also announced the updated "Dragon Ball Super Card Game" that starts with one starter deck, one special pack containing 4 booster packs and a promotional Vegeta card and a booster box with 24 packs. It was released on July 28, 2017. A line of six "Dragon Ball Super" Happy Meal toys were made available at Japanese McDonald's restaurants in May 2017. The average audience TV rating in Japan was 5.6% (Kanto region). The maximum audience rating was 8.4% (Episode 47) and the lowest rating was 3.5% (Episodes 109-110).

**Question**: When is episode 113 of dragon ball super coming out?

**Proposed Answer**: November 5, 2017

False

Table 9: Incorrect cases of self-verification. Generated texts are colored green.

I will provide ambiguous questions that can have multiple answers based on their different possible interpretations. Clarify the given question into several disambiguated questions and provide short factoid answers to each question. Subsequently, summarize them into a detailed long-form answer of at least three sentences. Here are some examples.

**Context**:
[1] Game of Thrones | Game of Thrones Game of Thrones is an American fantasy drama television series created by David Benioff and D. B. Weiss. ... and its seventh season ended on August 27, 2017. The series will conclude with its eighth season
[2] Game of Thrones | Game of Thrones is an American fantasy drama television series created by David Benioff and for HBO. It is an adaptation of "A Song of Ice and Fire", ... Set on the fictional continents of Westeros and Essos, "Game of Thrones" has a large ensemble cast
· · ·
[5] A Game of Thrones (comics) | A Game of Thrones (comics) A Game of Thrones is the comic book adaptation of George R. R. Martin's fantasy novel "A Game of Thrones", . . . It is intended to follow the story and atmosphere of the novel closely, at a rate of about a page of art for each page of text, and

**Question**: What kind of series is game of thrones?

**Disambiguations**:
DQ 1: What is the genre of the American television series Game of Thrones?
DA 1: fantasy drama
DQ 2: What is the genre of the comic book series A Game of Thrones?
DA 2: fantasy
· · ·
DQ 10: What is the genre of the board game A Game of Thrones?
DA 10: strategy

**Answer**: There are multiple works that share the title Game of Thrones. The first is a television series that is a fantasy drama, the second is a comic book series that is fantasy, the third is a book series that is fantasy, and the fourth is a board game that is a strategy game.

Table 10: Example prompt for the answer generation process. Generated texts are colored green.