# OpenReview forum: "Tree of Clarifications: Answering Ambiguous Questions with Retrieval-Augmented Large Language Models"
_EMNLP/2023/Conference — EMNLP 2023 Main_

### Official Review · Reviewer_GVet · 2023-08-04

**Soundness:** 4

**Excitement:**

4: Strong: This paper deepens the understanding of some phenomenon or lowers the barriers to an existing research direction.

**Paper Topic And Main Contributions:**

This paper introduces Tree of Clarification (ToC), which is a retrieval-based method that uses tree structure to clarify ambiguous questions. ToC method helps LLM to disambiguate questions, thereby achieving state-of-the-art result in ASQA dataset.

**Reasons To Accept:**

The proposed method, retrieval-augmented tree structure for clarification, is novel and convincing. Also, the reported result is solid.

**Reasons To Reject:**

The result is limited to a single dataset, ASQA. This work would be more convincing if it includes results on other datasets, to show that ToC is generalizable to other settings.

**Reproducibility:**

3: Could reproduce the results with some difficulty. The settings of parameters are underspecified or subjectively determined; the training/evaluation data are not widely available.

**Reviewer Confidence:**

3: Pretty sure, but there's a chance I missed something. Although I have a good feel for this area in general, I did not carefully check the paper's details, e.g., the math, experimental design, or novelty.

---

> ### Author Rebuttal · Authors · 2023-08-29
>
> We greatly appreciate the reviewer’s feedback. In particular, we are delighted that the reviewer considers our method as novel and convincing for the target task. We are also happy that the reviewer assesses the experimental results as robust.
>
> ***Response to Reason to Reject***
>
> 1. *The result is limited to a single dataset, ASQA. This work would be more convincing if it includes results on other datasets, to show that ToC is generalizable to other settings.*
>
> Although we could not provide enough discussion due to the short page limit, the key idea of ToC could be potentially generalized to the other tasks and model architectures. It has a model-agnostic structure that could effectively explore diverse paths of recursive reasoning, which would be helpful for tasks that require multiple reasoning, such as multi-hop QA.
> However, in this work, we focus on handling the ambiguity in open-domain QA by leveraging strong reasoning and the generation ability of recent LLM.  To the best of our knowledge, ASQA is the most recent and only benchmark that could evaluate long-form responses to the ambiguous question.
> We would continue to investigate the generalizability of ToC to diverse tasks, datasets, and LLMs for future work.

---

### Official Review · Reviewer_tfvE · 2023-08-05

**Soundness:** 4

**Excitement:**

4: Strong: This paper deepens the understanding of some phenomenon or lowers the barriers to an existing research direction.

**Paper Topic And Main Contributions:**

The paper introduces a novel framework called "Tree of Clarifications" (ToC) to address the issue of ambiguity in open-domain question answering. Ambiguous questions often have multiple interpretations, making it challenging to provide a comprehensive answer. ToC constructs a tree of disambiguations for the ambiguous question using few-shot prompting with external knowledge and uses it to generate a long-form answer that covers all possible interpretations. Experimental results show that ToC outperforms existing baselines on ASQA.

**Questions For The Authors:**

How do you ensure all interpretations are captured?

**Reasons To Accept:**

1. The paper introduces a novel framework called "Tree of Clarifications" (ToC) to address the challenge of ambiguity in open-domain question answering.
2. The experimental results demonstrate that ToC outperforms existing baselines on the ASQA benchmark in a few-shot setup across various metrics.
3. The proposed approach guides LLMs to explore diverse paths of clarification of the given AQ in a tree structure with the ability to prune unhelpful DQs. It is the first to combine retrieval systems with LLM for generating long-form answers to AQs.

**Reasons To Reject:**

The only limitation of this work is that it requires a tree of clarification process which makes the proposed solution complicated and it is a challenge to deploy such approach to production.

**Reproducibility:**

4: Could mostly reproduce the results, but there may be some variation because of sample variance or minor variations in their interpretation of the protocol or method.

**Reviewer Confidence:**

4: Quite sure. I tried to check the important points carefully. It's unlikely, though conceivable, that I missed something that should affect my ratings.

---

> ### Author Rebuttal · Authors · 2023-08-29
>
> We greatly appreciate your valuable comments. Specifically, we are delighted that you consider our method as a reasonable and novel solution for the challenge of ambiguity in open-domain QA. We are also pleased that you mentioned our experimental results strongly show the effectiveness of our method.
>
> ***Response to Reason to Reject***
>
> 1. *The only limitation of this work is that it requires a tree of clarification process which makes the proposed solution complicated and it is a challenge to deploy such approach to production.*
>
> Although ToC requires multiple generation processes by LLM, its maximum number is less than 10 per question. Exploration of the tree ends when it obtains 10 valid nodes and the clarification process generates 2-5 disambiguations for each question. Failing to expand three times in a row also terminates the exploration. Pruning steps consume a smaller amount of tokens since they only encode a single passage without few-shot exemplars.
>
> Compared to the existing ensemble methods such as self-consistency [1] which even cannot be directly adopted to the generative task, ToC achieves the state-of-the-art performance with a comparable number of LLM calls. It also outperforms the vanilla baseline with a single LLM call, significantly. For production, one could find the optimal way in such a trade-off between the clarification quality and computational cost.
>
> Regarding the concern about complexity, we will add these details to our final draft.
>
> [1] Wei et al., 2022, “Chain-of-Thought Prompting Elicits Reasoning in Large Language Models”

---

### Official Review · Reviewer_UCkq · 2023-08-12

**Soundness:** 3

**Excitement:**

4: Strong: This paper deepens the understanding of some phenomenon or lowers the barriers to an existing research direction.

**Paper Topic And Main Contributions:**

The authors introduce a recursive approach for clarifying ambiguous questions. Given an ambiguous question, relevant information is retrieved and an LLM is prompted to generate possible disambiguated questions. The process is repeated for the disambiguated questions. Answers are then combined to form a passage that answers the question, addressing ambiguity. The method demonstrates improvements over pure LLMs for ambiguous question answering.

Overall I think this is a very interesting idea, limited mainly by the lack of analysis on why ToC is effective.

**Questions For The Authors:**

- Do you think the improvements you get from RAC + TS come from adding the disambiguation tree, or from the fact that you are performing even more retrieval than with RAC alone? The biggest jump in the ablation seems to be the implicit RAG from RAC.
- Why not consider a diversity-based reranking of the retrieved passages, since your goal is to handle ambiguity?
- Why the different metrics between Table 2 and Table 3?
- Would this framework apply to other recursive question answering problems, such as multi-hop qa?


**Reasons To Accept:**

- Authors introduce a simple and sensible framework for disambiguating questions, where both question answering and question diambiguation are grounded with retrieved information.
- The paper achieves significant improvements over standard LLMs.
- Provides clear details on the prompts used for the RAC+TS framework.
- Includes useful ablations
- The paper contributes evidence that LLMs are effective at decomposing problems.

**Reasons To Reject:**

- Limited scope. Prompting and prompt chaining is a powerful framework, but mostly interesting when it enables general-purpose problem solving. This paper is limited to the ambiguous question problem.
- Source of improvements. Unclear whether the improvements can be attributed to the tree structure, or to the inclusion of more information via retrieval. The paper should include a comparison with retrieval-augmented methods.
- Missing a key analysis. While many of the experiments and comparisons are helpful to understand how ToC compares to existing approaches to ASQA, I'd like to see more analysis of the separate components of ToC. There's noo analysis on whether multiple levels of disambiguation are needed, how often multiple levels of disambiguation are necessary, or the LLMs ability to faithfully integrate multiple branches of the disambiguation tree, or the extent to which retrieval helps disambiguate (as opposed to the fact that retrieval clearly helps answer the questions). The most meaningful contribution is the notion of retrieval-augmented clarification, but I do not think the experiments isolate/evaluate RAC effectively.
- Missing latency/efficiency measurements. Using a recursive LLM framework comes with computational costs; this should be measured by the authors. It's difficult when using an API, but even a sense of the typical numbers of API calls would be helpful.
- It seems like no details for standard PaLM and GPT prompting were included. The prompt examples seem to all be for the RAC+TS prompting.

**Reproducibility:**

4: Could mostly reproduce the results, but there may be some variation because of sample variance or minor variations in their interpretation of the protocol or method.

**Reviewer Confidence:**

4: Quite sure. I tried to check the important points carefully. It's unlikely, though conceivable, that I missed something that should affect my ratings.

---

> ### Author Rebuttal · Authors · 2023-08-29
>
> We greatly appreciate the reviewer’s detailed and helpful feedback. In particular, we are delighted that the reviewer agrees the proposed method is straightforward and sensible for the target task. We are also happy that the reviewer assesses the paper as clear and detailed and confirms our experiments demonstrate the effectiveness of the method and show helpful ablation studies.
>
> ***Response to Reasons to Reject***
>
> 1. *Limited scope. Prompting and prompt chaining is a powerful framework, but mostly interesting when it enables general-purpose problem solving. This paper is limited to the ambiguous question problem.*
>
> Although we could not provide enough discussion due to the short page limit, the key idea of ToC could be potentially generalized to the other tasks and model architectures. It has a model-agnostic structure that could effectively explore diverse paths of recursive reasoning, which would be helpful for tasks that require multiple reasoning, such as multi-hop QA. \
> In this work, we focus on one of the most common and challenging issues, the ambiguity in open-domain QA. Our work holds significance in that it tackles the crucial issue where the prompting methods for general-purpose such as chain-of-thoughts or self-consistency [1] were ineffective or not applicable.
>
> 2. *Source of improvements. Unclear whether the improvements can be attributed to the tree structure, or to the inclusion of more information via retrieval. The paper should include a comparison with retrieval-augmented methods.*
>
> In Table 1, we demonstrate how much retrieval contributes to the improvement by isolating RAC and provide the comparison with other retrieval-augmented methods, “T5-large w/ JPR” from the previous work [3]. In our pilot experiment, we found that simply increasing the number of retrieved passages does not improve the performance, and then fixed the number of passages as 5. On the other hand, tree structure further enhances the retrieval diversity and relevance, enabling context-specific search for each interpretation. Unlike prior studies' efforts to improve recall scores in retrieval tasks, TS is designed especially to broaden the coverage of retrieved passages.
>
> 3. *Missing a key analysis*
>   - *Whether multiple levels of disambiguation are needed*
>
> The tree structure (TS) is motivated by the previous work [4] that reports a single ambiguous question may pose multiple types of ambiguity. Moreover, improvement from the pruning method could also implicate the effectiveness of using multiple levels. TS with pruning can explore the deeper level of the tree while the TS-only baseline reaches the maximum node limit earlier in the shallower level. In Table 1, TS with pruning shows better performance in F1 and DR scores than the TS-only baseline.
>
>   - *how often multiple levels of disambiguation are necessary*
>
> Thanks for the great suggestion but it would be not trivial to precisely measure it with an analysis. We will try to handle it by providing statistics about the number of nodes or depth in resulting trees and reporting them in our final draft.
>
>   - *LLMs ability to faithfully integrate multiple branches of the disambiguation tree*
>
> In Table 3, we provide the accuracy of the disambiguation trees. With our pruning technique, intermediate trees are enhanced, leading to the improvement of the final answer. It would be one of the pieces of evidence showing that the answer generation module faithfully refers to the disambiguation trees. When removing the intermediate tree from the input prompt, its performance largely degrades.
>
>   - *the extent to which retrieval helps disambiguate (as opposed to the fact that retrieval clearly helps answer the questions). The most meaningful contribution is the notion of retrieval-augmented clarification, but I do not think the experiments isolate/evaluate RAC effectively.*
>
> In Table 1, we demonstrate how much retrieval contributes to the improvement by isolating RAC from TS. RAC largely advances the model performance and TS further enhances it.
>
> 4. *Missing latency/efficiency measurements. Using a recursive LLM framework comes with computational costs; this should be measured by the authors. It's difficult when using an API, but even a sense of the typical numbers of API calls would be helpful.*
>
> Although ToC requires the multiple generation process by LLM, its maximum number is less than 10 per question. Exploration of the tree ends when it obtains 10 valid nodes and the clarification process generates 2-5 disambiguations for each question. Failing to expand three times in a row also terminates the exploration. Pruning steps consume a smaller amount of tokens since they encode a single passage without few-shot exemplars. Compared to the existing ensemble methods such as self-consistency [1] which even cannot be directly adopted to the generative task, ToC achieves the state-of-the-art performance with a comparable number of LLM calls.\
> Regarding the concern about complexity, we will add these details to our final draft.
>
> 5. *It seems like no details for standard PaLM and GPT prompting were included. The prompt examples seem to all be for the RAC+TS prompting.*
>
> In Table 1, the performances for standard PaLM and GPT without retrieval are reported by the previous work [2] so please refer to it for the prompts. For our closed-book setup in Table 2, we use the same prompt as that in Table 6 except “Context” and “Disambiguations” sections.
>
> ***Answers to Questions***
> - Q1: *Do you think the improvements you get from RAC + TS come from adding the disambiguation tree, or from the fact that you are performing even more retrieval than with RAC alone? The biggest jump in the ablation seems to be the implicit RAG from RAC.*
>
> As mentioned in Response 2 (*Source of improvements*), simply increasing the number of passages is ineffective. It could introduce noisy and redundant distractors for the limited input length of LLM, as reported by recent works [5,6]. Repeatedly performing retrieval with similar queries would not expand the coverage of the passage set. On the other hand, tree structure enhances the diversity and coverage of retrieved passages. It would be the key advantage of ToC and where the most improvement comes from.
>
> - Q2: *Why not consider a diversity-based reranking of the retrieved passages, since your goal is to handle ambiguity?*
>
> In each node, passages are reranked regarding each DQ, which enhances the diversity of passages. Thanks for the interesting idea. Adopting diversity-specific techniques might improve the performance.
>
> - Q3: *Why the different metrics between Table 2 and Table 3?*
>
> We use D-F1 to evaluate the long-form answers to the ambiguous questions. In Table 3, we show the accuracy of intermediate pairs of disambiguated questions and their short answers. “Answer-F1” is used to validate how accurate the short answers are. Appendix A.2 provides the details about it. We will add a footnote for it.
>
> - Q4: *Would this framework apply to other recursive question answering problems, such as multi-hop qa?*
>
> For handling multi-hop QA tasks, current studies on the chain of reasoning are introduced and they could be viewed as a special case of the tree structure with depth-first search. Tree structure might be applied to compare and validate multiple reasoning chains. Thanks for the interesting idea.
>
> [1] Wei et al., 2022, “Chain-of-Thought Prompting Elicits Reasoning in Large Language Models”\
> [2] Amplayo et al., ACL 2023, “Query Refinement Prompts for Closed-Book Long-Form QA”\
> [3] Stelmakh et al., EMNLP 2022, "Asqa: Factoid questions meet long-form answers."\
> [4] Min et al., EMNLP 2020, “AmbigQA: Answering Ambiguous Open-domain Questions”\
> [5] Shi et al., 2023, “REPLUG: Retrieval-Augmented Black-Box Language Models”\
> [6] BehnamGahder et al., 2023, “Can Retriever-Augmented Language Models Reason? The Blame Game Between the Retriever and the Language Model”

---

### Official Review · Reviewer_T734 · 2023-08-12

**Soundness:** 4

**Excitement:**

4: Strong: This paper deepens the understanding of some phenomenon or lowers the barriers to an existing research direction.

**Paper Topic And Main Contributions:**

The paper introduces a framework called "Tree of Clarifications" (TOC) for answering ambiguous open-domain questions where they recursively construct a tree to explore different interpretations (disambiguations) of the ambiguous question. TOC uses retrieval to find relevant passages and augment them to prompts for LLMs. It prunes unhelpful or irrelevant disambiguations using a self-verification method. The experiments on the ASQA benchmark show TOC outperforms previous baselines and fully supervised models trained on the whole dataset.


**Reasons To Accept:**

The paper is well written. The novel idea of using a recursive tree structure to methodically explore ambiguities seems like a simple and reasonable method for long-form ambiguous QA. The paper also does important ablations showing the contributions of different components of their framework. different Strong Results: TOC Outperforms fully-supervised models with only 5 examples, and achieves new SOTA results on the ASQA benchmark.


**Reasons To Reject:**

The paper only presents experiments with only one LLM model (GPT-3) on one dataset (ASQA).

**Reproducibility:**

4: Could mostly reproduce the results, but there may be some variation because of sample variance or minor variations in their interpretation of the protocol or method.

**Reviewer Confidence:**

3: Pretty sure, but there's a chance I missed something. Although I have a good feel for this area in general, I did not carefully check the paper's details, e.g., the math, experimental design, or novelty.

---

> ### Author Rebuttal · Authors · 2023-08-29
>
> We greatly appreciate the reviewer’s valuable comments. In particular, we are delighted that the reviewer agrees the proposed method is straightforward and reasonable for the target task. We are also happy that the reviewer assesses the paper as well-written and confirms our experiments demonstrate the effectiveness of the method and show the helpful ablation studies.
>
> ***Response to Reason to Reject***
>
> 1. *The paper only presents experiments with only one LLM model (GPT-3) on one dataset (ASQA).*
>
> Although we could not provide enough discussion due to the short page limit, the key idea of ToC could be potentially generalized to the other tasks and model architectures. It has a model-agnostic structure that could effectively explore diverse paths of recursive reasoning, which would be helpful for tasks that require multiple reasoning, such as multi-hop QA.
>
>   However, in this work, we focus on handling the ambiguity in open-domain QA by leveraging strong reasoning and generation ability of recent LLMs.
> - Dataset: To the best of our knowledge, ASQA [1] is the most recent and only benchmark that evaluates long-form responses to the ambiguous questions.
> - Model: For comparing our closed-book baseline, we chose the same architecture (i.e., GPT-3 text-davinci-002) with the previous work [2], except for the unreleased model, PaLM 1 540B. Fine-tuning the smaller scale of LMs (e.g., T5, LLaMa, Vicuna, and MPT) would be less effective due to the small training set as the baselines show.
>
> We would continue to investigate the generalizability of ToC to diverse tasks, datasets, and LLMs for future work.
>
> [1] Stelmakh et al., EMNLP 2022, "ASQA: Factoid questions meet long-form answers."\
> [2] Amplayo et al., ACL 2023, “Query Refinement Prompts for Closed-Book Long-Form QA”

---

### Meta-Review · Senior_Area_Chairs · 2023-09-18

**Recommendation:** 5

**Metareview:**

The authors propose to tackle ambiguous open-domain questions by iteratively building a tree of clarifications, which holds interpretations for the original question. Then, the long-form answer is generated conditioned on these interpretations.

Reviewers all like the proposed approach for the problem, and are impressed with the improvements over the baselines.
Some reviewers are favorable towards the ablations in the paper but reviewer UCkq mentions that analysis is missing w.r.t the different components of the system.
Reviewers also mention that the latency slowdown of using the proposed approach is not analyzed, and that the authors only experiment with one dataset.

Improvement: I noted that you use SentenceBERT for ranking answer sentences. The specific research area is called Answer Sentence Selection, and you can find the state of the art below:

Siddhant Garg, Thuy Vu, Alessandro Moschitti. TANDA: Transfer and Adapt Pre-Trained Transformer Models for Answer Sentence Selection. AAAI 2020.

Ivano Lauriola and Alessandro Moschitti: "Answer sentence selection using local and global context in transformer models", ECIR 2021.

---

### Decision · Program_Chairs · 2023-10-07

**Decision:**

Accept-Main

**Comment:**

The authors propose to tackle ambiguous open-domain questions by iteratively building a tree of clarifications, which holds interpretations for the original question. Then, the long-form answer is generated conditioned on these interpretations.

Reviewers all like the proposed approach for the problem, and are impressed with the improvements over the baselines.
Some reviewers are favorable towards the ablations in the paper but reviewer UCkq mentions that analysis is missing w.r.t the different components of the system.
Reviewers also mention that the latency slowdown of using the proposed approach is not analyzed, and that the authors only experiment with one dataset.

Improvement: I noted that you use SentenceBERT for ranking answer sentences. The specific research area is called Answer Sentence Selection, and you can find the state of the art below:

Siddhant Garg, Thuy Vu, Alessandro Moschitti. TANDA: Transfer and Adapt Pre-Trained Transformer Models for Answer Sentence Selection. AAAI 2020.

Ivano Lauriola and Alessandro Moschitti: "Answer sentence selection using local and global context in transformer models", ECIR 2021.